



**Exogenous phosphorus compounds interact with nitrogen availability to regulate dynamics of soil inorganic phosphorus fractions in a meadow steppe**

Heyong Liu [a,b], Ruzhen Wang [b,*], Hongyi Wang [b], Yanzhuo Cao [b,c], Feike A. Dijkstra [d], Zhan Shi [b], Jiangping Cai [b], Zhengwen Wang [b], Hongtao Zou [a,*], Yong Jiang [b]

[a] *College of Land and Environment, Shenyang Agricultural University, Shenyang 110866, China*

[b] *Erguna Forest-Steppe Ecotone Ecosystem Research Station, Institute of Applied Ecology, Chinese Academy of Sciences, Shenyang 110016, China*

[c] *Key Laboratory of Regional Environment and Eco-Remediation, College of*

*Environment, Shenyang University, Shenyang 110044, China*

[d] *Centre for Carbon, Water and Food, School of Life and Environmental Sciences, The University of Sydney, Sydney, NSW 2006, Australia*

[*] Correspondence to:

Dr. Ruzhen Wang, Tel: +86 24 83970603; *E–mail address:* ruzhenwang@iae.ac.cn

Dr. Hongtao Zou, Tel: +86 24 88487155; *E–mail address:* zouhongtao2001@163.com





**Abstract**

Here we investigated the effects of P compounds ($KH_2PO_4$ and $Ca(H_2PO_4)_2$) with different addition rates of 0, 20, 40, 60, 80 and 100 kg P ha$^{-1}$ yr$^{-1}$ and $NH_4NO_3$ addition (0 and 100 kg N ha$^{-1}$ yr$^{-1}$) on soil labile (Olsen-P), moderate-cycling and

recalcitrant inorganic phosphorus (IP) fractions in a calcareous grassland of northeastern China. Soil moderate-cycling IP, not readily available to plants but transforming into available P quickly, include variscite (Al-P), strengite (Fe-P), dicalcium phosphate ($Ca_2$-P) and octacalcium phosphate ($Ca_8$-P); recalcitrant fractions include hydroxylapatite ($Ca_{10}$-P) and occluded P (O-P). Soil labile and

moderate-cycling IP fractions and total P significantly increased with increasing P addition rates, with higher concentrations detected for $KH_2PO_4$ than for $Ca(H_2PO_4)_2$ addition. Combined N and P treatments showed lower moderate-cycling IP fractions compared to ambient N conditions due to enhanced plant productivity. Moderate-cycling IP was mainly regulated by aboveground plant biomass with

$KH_2PO_4$ addition, while by soil pH and plant biomass with addition of $Ca(H_2PO_4)_2$. $Ca(H_2PO_4)_2$ addition significantly increased the soil recalcitrant IP ($Ca_{10}$-P) fraction, while $KH_2PO_4$ addition showed no impact on it. A significant positive correlation was detected between soil moderate-cycling IP fractions and soil Olsen-P which illustrated that moderate-cycling IP fractions were important sources for available P. Our results

suggest that moderate-cycling IP fractions are essential for grassland P biogeochemical cycling and chemical form of P fertilizer should be considered during fertilization management for maintaining soil available P.

**Key words** semi-arid steppe; soil phosphorus; fertilization management;

mineral-bound phosphorus; occluded phosphorus

## 1. Introduction

Phosphorus (P) is an essential nutrient affecting terrestrial plant productivity and ecosystem functions (Luo et al., 2015). Soil inorganic P (IP) could occupy 50-90% of soil total P (Jiang and Gu, 1989; Feng et al., 2016) and mainly includes adsorbed and

mineral forms of phosphate (Hinsinger, 2001). Soil mineral-bound phosphate falls into two groups: those containing calcium (Ca-P) and those containing iron and aluminum ( Chen et al., 2002), which are the leading sources of plant available P (Meena et al., 2018). The detailed classification of these two groups includes variscite (Al-P), strengite (Fe-P), dicalcium phosphate ($Ca_2$-P), octacalcium phosphate ($Ca_8$-P),

hydroxylapatite ($Ca_{10}$-P) and occluded P (O-P) (Zhao et al., 2019). Soil IP is dominated by Al-P and Fe-P in acidic soils and by Ca-P fractions in calcareous soils (Baumann et al., 2017). The solubility of soil Ca-P fractions decreases in the order $Ca_2$-P > $Ca_8$-P > $Ca_{10}$-P (Jiang and Gu, 1989). Although these element-bound P fractions are not readily available to plants, Al-P, Fe-P, $Ca_2$-P and $Ca_8$-P can be

converted to free phosphates serving as important buffering pools for available P (Herlihy and McGrath, 2006; Zhao et al., 2019). Based on P transforming dynamics, soil IP could be fractionated into labile/ available P (Olsen P), moderate-cycling P fractions (Al-P, Fe-P, $Ca_2$-P and $Ca_8$-P) and recalcitrant P fractions ($Ca_{10}$-P and O-P). But the category is not universal which mainly depends on the sensitivity of P

fractions in different studies (Schrijver et al., 2012 *vs.* Zhao et al., 2019). Therefore, understanding soil IP transformation is crucial for nutrient cycling in terrestrial ecosystems.

Adding P to soil is an effective way to enhance soil available P and sustain plant productivity (Marklein and Houlton, 2012). However, P addition changes ecosystem P

cycling (Herlihy and McGrath, 2006) and the reactions between free phosphate and



cations in soil thereby influencing the content and transformation of soil IP fractions (Walker and Syers, 1976). As such, chronic P input can result in insoluble phosphate accumulation and decrease P use efficiency without providing additional benefits for plant growth (Maistry et al., 2015). This could then increase the risk of soil P leaching

losses (Shen et al., 2004). Therefore, exploring the effect of P addition on soil IP fractions is important for understanding soil available P supply and P dynamics in terrestrial ecosystems (Sigua et al., 2017).

Biogeochemical P dynamics can also be influenced by the chemical form of applied P (Margenot et al., 2017). Frequently-used P compounds include $Ca(H_2PO_4)_2$

(slow-release P fertilizer) and $KH_2PO_4$ (fast-release P fertilizer) (Mortvedt et al., 1999), which have played critial roles in agricultural ecosystems (Nguyen et al., 2012). Both P fertilizers can convert into various inorganic P fractions including Fe-P (especially in paddy soils with high $Fe^{2+}$) (Sah et al., 1989) and Al-P, which can account for 33% of the total soil P in weathered soils (Margenot et al., 2017). And

$Ca_2$-P and $Ca_8$-P could be also formed with P addition, especially in calcareous soils with pH higher than 7 (Al-Khateeb et al., 1986). In a previous study it was shown that long-term application of $Ca(H_2PO_4)_2$ increased soil inorganic P and total P (TP) concentrations in a calcareous soil (Wang et al. 2010). However, long-term $Ca(H_2PO_4)_2$ addition decreased P availability due to fixation of P to the soil

constituents (Xiong et al., 2018). $KH_2PO_4$ addition was suggested to enhance the reaction of phosphate with Fe and Al oxides and with hydroxyl ions under lower soil pH conditions (Shuman, 1988). While these studies provided insight into the relationships between P addition and IP fractions (Condron and Goh, 1989), little is known about the effects of different P compound additions on soil IP fractions and P

availability.



Nitrogen enrichment can influence soil chemical properties thereby affecting soil P transformations, and consequently above-ground biomass and foliar P concentrations (Crowley et al., 2012). For instance, N addition often decreases soil pH and promotes the release of P from Fe and Al oxides (Gustafsson et al., 2012;Yang et

al., 2014), and can result in redistribution of soil IP fractions (Zhao et al., 2019). Nitrogen addition could also promote the release of P from soil organic P by increasing phosphatase activities (Olander and Vitousek, 2000). The effect of combined N and P addition on ecosystem P dynamics may further depend on their relative amounts added to the soil. For instance, inorganic P solubility and availability

were particularly large when P was applied with N (Ross et al., 1995), while N and P applied at higher N:P ratios increased plant P use efficiency and leaf phosphatase activity (Hogan et al., 2010). It is well known that variation in N:P input ratios can also alter plant litter N:P ratios (Vitousek et al., 2010; Güsewell, 2004; Sun et al., 2018), thereby further affecting N and P availability through litter decomposition.

However, it remains unclear how applications of N and P added at different ratios will affect soil IP fractions.

       The meadow steppe in northern China is an important part of the Eurasian grassland ecosystem (Kang et al., 2007). With the intensive environmental stresses of overgrazing and enhanced outputs of animal products from a sharply rising population,

Inner Mongolia grassland ecosystems have potentially become more nutrient-limited, predominantly by N and P (Kang et al., 2007; Gong et al., 2011). Hence, N and P additions are necessary to enhance ecosystem productivity in the meadow steppe. The purpose of our study was to monitor the effects of various levels of $KH_2PO_4$ and $Ca(H_2PO_4)_2$ with and without N addition on concentrations of soil IP fractions,

available P and TP. We hypothesized that 1) soil labile IP (Olsen-P) and



moderate-cycling IP fractions (Al-P, Fe-P, $Ca_2$-P and $Ca_8$-P) would significantly

increase with $KH_2PO_4$ and $Ca(H_2PO_4)_2$ addition, while soil recalcitrant IP fractions

($Ca_{10}$-P and O-P) would not be affected, because of different solubility in IP fractions;

2) addition of soluble $KH_2PO_4$ would be more beneficial to increasing soil labile IP,

moderate-cycling IP fractions, recalcitrant P fractions and total P (TP) than

less-soluble $Ca(H_2PO_4)_2$, because of faster conversion into these fractions with the

more soluble $KH_2PO_4$; 3) the concentrations of soil IP fractions would be lower under

combined P and N additions than that under P addition alone due to facilitation of

plant P uptake with N addition.

## 2. Materials and methods

### 2.1. Study sites and experimental design

The study site (119º 22′ E, 50º 10′ N, elevation 523 m a.s.l.) is located in the Erguna

Forest-Steppe Ecotone Research Station of Inner Mongolia, China. This area belongs

to a temperate continental monsoon climate. The mean annual precipitation and mean

annual temperature is 375 mm and -3 ℃, respectively. Rainfall of the site is mainly

concentrated during the period from June to August and the average growing season is

about 150 days. The soil is a Chernozem according to the Food and Agricultural

Organization of the United Nations classification (WRB, 2014), and the soil chemical

characteristics of the site are reported in Table 1. The dominant plant species include

*Stipa baicalensis*, *Leymus chinensis* and *Carex duriuscula*.

The experiment, established in 2014, was arranged in a randomized block design

with 24 treatments and five replicates. Phosphorus addition included two compounds,

*i.e.,* $KH_2PO_4$ and $Ca(H_2PO_4)_2$, and were applied at six levels: 0, 20, 40, 60, 80 and 100

kg P $ha^{-1}$ $yr^{-1}$. Half of the plots were applied with 100 kg N $ha^{-1}$ $yr^{-1}$ in the form of



$NH_4NO_3$. Phosphorus and N were added in the middle of May. All treatment plots

were balanced for K using potassium chloride (KCl) to maintain the same amount of

K input as in the treatment with 100 kg P ha$^{-1}$ yr$^{-1}$ of $KH_2PO_4$ addition (132 kg K ha$^{-1}$).

To balance for Cl along with KCl addition, $CaCl_2$ was applied to maintain equal Cl

inputs for all plots (121 kg Cl ha$^{-1}$). Calcium was not balanced in this calcareous soil,

where Ca is already abundant. In this study we chose the same control plots for the

$KH_2PO_4$ and $Ca(H_2PO_4)_2$ treatment, so that in total, there were 110 plots of 8 m ×8 m,

separated by 1-m wide buffer zones. The initial pH of $KH_2PO_4$ and $Ca(H_2PO_4)_2$

solutions were 4.91 and 4.50, respectively.

**2.2. Sample collection**

In August 2016, aboveground biomass was harvested by clipping all living tissues

using a 1 m ×1 m quadrat placed randomly within each plot. The plants were sorted

to species and oven-dried at 65 ℃ for 48 h. Soil samples were taken from the surface

(10 cm depth) using a 5-cm diameter soil auger. Five soil cores were collected from

each plot and mixed into one composite sample. Each sample was air-dried and

passed through a 2-mm sieve to remove litter and detritus. A subsample of the

air-dried soil was ground using a ball mill to pass through a 0.15-mm sieve for further

analyses of IP fractions and TP.

    **2.3. Soil chemical analyses**

Soil pH was measured in a 1:5 soil-to-water slurry with a pH meter (S210

SevenCompact™, Mettler, Germany). Soil IP fractions were extracted according to

the sequential fractionation scheme proposed by Chang and Jackson (1957) which is

modified to suit for calcareous soils (Jiang and Gu, 1989). Briefly, $Ca_2$-P was

determined by shaking 0.5g soil with 25 ml 0.25 M $NaHCO_3$ (pH 7.5), and then

centrifuged at 3500 rpm for 8 min to measure soil $Ca_2$-P. The remaining soil was

washed two times with 25 ml 95% $C_2H_5OH$ and extracted with 25 ml 0.5 M $NH_4Ac$

(pH = 4.2) to determine soil $Ca_8$-P (Jiang and Gu, 1989). After this, the soil was

shaken with 25 ml 1 M $NH_4Cl$ and centrifuged at 3500 rpm to discard the supernatant.

The remaining soil was then shaken with 25 ml 0.5 M $NH_4F$ (pH 8.2) and centrifuged

at 3500 rpm for 8 min to analyze the Al-P fraction in the supernatant. The remaining

soil was washed two times with 25 ml saturated NaCl, and then sequentially shaken

with 25 ml mixture of 0.1 M NaOH and $Na_2CO_3$ for 2 h at 25 ℃, and then centrifuged

at 4500 rpm for 10 min to measure Fe-P in the supernatant. The remaining samples

were washed as above and extracted with 25 ml mixture of 0.3 M

$Na_3$(citrate)-$Na_2S_2O_4$ and 0.5 M NaOH to measure O-P. Finally, the remaining

samples were shaken with 0.25 M $H_2SO_4$ for 1 h at 25 ℃, and centrifuged at 3800

rpm for 10 min to determine $Ca_{10}$-P. The $Ca_2$-P and $Ca_8$-P fractions were extracted

separately from the Al-P, Fe-P, O-P and $Ca_{10}$-P frations. The P concentration in all the

extractants was determined by the molybdenum blue colorimetric method at

wavelength of 700 nm with a UV-VIS spectrometer (UV-1700, Shimazu) (Murphy

and Riley, 1962). Total IP (TIP) concentration was defined as the sum of

moderate-cycling IP (Al-P, Fe-P, $Ca_2$-P and $Ca_8$-P) and recalcitrant IP ($Ca_{10}$-P and

O-P). The potential limitation of the extraction methods is that they may not be very

specific in separating the different forms of P minerals, as a small amount of other

phosphate dissolved in the extractants (Jiang and Gu, 1989).

Soil TP concentration was determined after digestion with 8 ml 85% $HNO_3$ +4 ml

72% $HClO_4$ +1 ml 40% HF (Sommers and Nelson, 1972), and Olsen-P was extracted

from air-dried soil with 0.5 M $NaHCO_3$ (pH 8.5) (Olsen et al., 1954). Plant TP

concentration was determined by acid digestion using $H_2SO_4$-$H_2O_2$ (Thomas et al.,

1967). Soil TP, Olsen P and plant TP were analyzed by the molybdenum blue



colorimetric method at 700 nm.

**2.4. Statistical analyses**

Plant P uptake of three dominant species was calculated using the following

equation:

$$P\ uptake = \sum_{i}^{n} P_i \times B_i,$$

where $P_i$ is TP concentration of species $i$, and $B_i$ is the biomass of species $i$. All the

data were shown as mean ± standard error. The Kolmogorov-Smirnov test was

performed to determine whether data had a normal distribution. Three-way ANOVAs

were conducted to determine the effects of N addition (N), P addition rate ($P_r$), P

compounds ($P_t$) and their interactions on soil IP fractions and Olsen-P concentration.

For each P compound and N treatment, the effect of P addition rates on

moderate-cycling IP fractions and plant P were determined using polynomial

contrasts, the effect of P addition rates on plant biomass, soil pH, recalcitrant IP

fractions, Olsen-P and TP were analyzed using Duncan's multiple range tests.

Student t-test was used to determine the difference between two P compounds within

each P addition rate and N treatment (without and with N) and between N treatments

within each P compound and addition rate. For moderate-cycling IP fractions,

one-way analysis of covariance (ANCOVA) was employed to distinguish the slopes

between the two N treatments (without N *vs.* with N) for each P compound and

between the two P compounds ($KH_2PO_4$ *vs.* $Ca(H_2PO_4)_2$) for each N treatment.

Pearson correlations were used to test the relationships between soil variables. All

the above statistics were carried out using SPSS 16.0 (SPSS Inc., Chicago, USA).

Structural equation models (SEM) were built to clarify direct and indirect

relationships between plant biomass, soil pH, moderate-cycling IP fractions,

recalcitrant IP fraction, Olsen-P and TP. Chi-square test, Akaike information criteria





(AIC) and the root mean square error of approximation (RMSEA) were used to

evaluate the fit of the model. The SEM analyses were performed using AMOS 7.0

(Amos Development Co., Greene, Maine, USA). Statistical significance was

accepted at $P < 0.05$.

## 3. Results

### 3.1. Aboveground plant biomass and soil pH

Nitrogen addition significantly increased aboveground biomass production (Fig. 1a,b).

Aboveground biomass production did not show a clear trend in response to different

levels of P addition, either as $KH_2PO_4$ or as $Ca(H_2PO_4)_2$ (Fig. 1a,b). Nitrogen addition

significantly increased plant P uptake of the three dominant species *Stipa baicalensis*,

*Leymus chinensis* and *Carex duriuscula* for both P compounds (Fig. 1c,d). Plant P

uptake increased with increasing P addition rate with significantly higher overall

$KH_2PO_4$ effect than $Ca(H_2PO_4)_2$. $KH_2PO_4$ addition showed no impact on soil pH (Fig.

1e), while $Ca(H_2PO_4)_2$ decreased soil pH without N addition (Fig. 1f).

### 3.2. Soil moderate-cycling inorganic phosphorus fractions

For both $KH_2PO_4$ and $Ca(H_2PO_4)_2$ additions, Al-P and Fe-P concentrations

significantly increased with increased P addition rates (Fig. 2a, b, c, d). Al-P and Fe-P

concentrations were higher with $KH_2PO_4$ than with $Ca(H_2PO_4)_2$ addition, especially at

higher P rates (Fig. 2; Table S1). This resulted in significant interactive $P_t \times P_r$ effects

on Al-P and Fe-P concentrations (Table 2). For instance, Al-P was higher with

$KH_2PO_4$ addition than $Ca(H_2PO_4)_2$ at 60 kg P ha$^{-1}$ yr$^{-1}$ without N addition and at 60,

80 and 100 kg P ha$^{-1}$ yr$^{-1}$ levels with N addition. Nitrogen addition decreased both

Al-P and Fe-P concentrations, particularly at higher levels of P addition (Fig. 2; Table

25    2 and S1).





Addition of both P compounds significantly increased $Ca_2$-P and $Ca_8$-P

concentrations with and without N addition (Fig. 3a, b, c, d). With $KH_2PO_4$ addition,

soil $Ca_2$-P and $Ca_8$-P concentrations were higher than with $Ca(H_2PO_4)_2$ addition (Fig.

3; Table S1). Nitrogen addition significantly decreased soil $Ca_2$-P and $Ca_8$-P

concentrations for both P compounds in some P addition levels (Fig. 3; Table S1).

Therefore, significant $P_t \times P_r$ interactive effects were detected on $Ca_2$-P and $Ca_8$-P and

$P_r \times N$ interactive effect on $Ca_2$-P (Table 2).

### 3.3. Soil recalcitrant inorganic phosphorus fractions

$Ca(H_2PO_4)_2$ addition significantly increased soil $Ca_{10}$-P concentration, while $KH_2PO_4$

addition showed no impact (Fig. 4a, b). $Ca_{10}$-P concentration was higher with

$Ca(H_2PO_4)_2$ addition than with $KH_2PO_4$ addition for all levels except for 60 kg P ha$^{-1}$

yr$^{-1}$ without N addition and for 0, 40 and 100 kg P ha$^{-1}$ yr$^{-1}$ with N addition (Fig. 4a, b;

Table S2). There was no main N addition and $N \times P_r$ interactive effect on $Ca_{10}$-P for

both P compounds (Table 2). Nitrogen addition had also no significant effect on O-P

with $KH_2PO_4$ addition but significantly decreased it with $Ca(H_2PO_4)_2$ addition at 60

and 80 kg P ha$^{-1}$ yr$^{-1}$ (Fig. 4c, d). Soil O-P showed a hump-shaped relationship along

the P addition gradient with $KH_2PO_4$ when added with N, and with $Ca(H_2PO_4)_2$

independent of N addition. The O-P concentration was lower with $Ca(H_2PO_4)_2$

addition than with $KH_2PO_4$ addition for 60 and 80 kg P ha$^{-1}$ yr$^{-1}$ with N addition (Fig.

4c, d; Table S2). The relative proportions of O-P and $Ca_{10}$-P to TIP decreased while

the proportions of Al-P, $Ca_2$-P, and $Ca_8$-P increased with increasing P addition rate for

both P compounds (Fig. S1).

### 3.3. Soil Olsen-P and total P

For both P compounds, P addition significantly increased Olsen-P concentration

regardless of N addition (Fig.5a, b). Olsen-P concentration increased more strongly

with increased levels of $KH_2PO_4$ addition than with $Ca(H_2PO_4)_2$ addition, resulting in significantly higher Olsen-P with $KH_2PO_4$ addition in the 100 kg P ha$^{-1}$ yr$^{-1}$ treatment with and without N addition (Fig. 5a, b). Soil TP increased with P addition both with $KH_2PO_4$ and $Ca(H_2PO_4)_2$ addition without and with N addition (Fig. 5c, d).

**3.4. Correlation between soil inorganic fractions with soil characteristics**

For both P compounds, Al-P, Fe-P, $Ca_2$-P and $Ca_8$-P had significantly positive correlations with each other. In addition, TP and Olsen-P were all positively correlated with Al-P, Fe-P, $Ca_2$-P, and $Ca_8$-P ($P < 0.01$) (Table 3). $Ca_{10}$-P had significantly positive correlations with Fe-P and $Ca_8$-P for both P compounds (Table 3).

Furthermore, soil TIP concentration was positively correlated with the level of P addition for both P compounds (Fig. S2).

The SEM suggested that plant biomass had a negative impact on moderate-cycling IP both with $KH_2PO_4$ (Fig. 6a) and with $Ca(H_2PO_4)_2$ addition (Fig. 6b), while soil pH also negatively influenced moderate-cycling IP with $Ca(H_2PO_4)_2$. Moderate-cycling IP

fractions drove the increase in Olsen-P (labile P) concentration for both P compounds, with a higher contribution for $KH_2PO_4$ addition (Fig. 6a,b). Olsen-P was negatively affected by recalcitrant IP only with $Ca(H_2PO_4)_2$ addition (Fig. 6b).

**4. Discussion**

**4.1. Effect of P additions rates on soil IP fractions**

With increasing P addition levels, the increase in soil Olsen-P and moderate-cycling IP fractions was consistent with our first hypothesis and the findings from previous studies (Wang et al., 2010; Zhao et al., 2019). Soil IP fractions can be affected by nutrient addition, soil type and soil chemical properties (Daly et al., 2001; Stroia et al.,

2011). The positive linear correlation of TIP concentration and P addition rates (Fig.



S2) indicates that applied P was immobilized mainly into inorganic forms (Chauhan et al., 1981) and converted to various P fractions (Piegholdt et al., 2013). Most of the applied P transformed into moderate-cycling IP fractions as seen from the decrease in the relative proportion of recalcitrant IP fractions (Fig. S1). Indeed, applied P fertilizer

can be quickly bound by P-fixing constituents, *e.g.*, Fe/Al oxides and clay minerals (Devau et al., 2011). In calcareous soils, precipitation is the main process retaining applied P in soils, especially precipitation with Ca at relatively high soil pH (Wang et al., 2010). Likely, the amount of P from input processes (exogenous P and weathering) was higher than the amount from output processes (plant uptake, erosion and leaching

losses), which can then cause P accumulation in the soil as insoluble P fractions (Song et al., 2017). Therefore, both the monopotassium phosphate and monocalcium phosphate fertilizers transformed into moderate-cycling IP fractions and contributed to the increase in Al-P, Fe-P, $Ca_2$-P and $Ca_8$-P concentrations (Fig. 7).

Inconsistent with our first hypothesis, we found significant changes in recalcitrant

P fractions ($Ca_{10}$-P and O-P), which accounted for 21%-73% of TIP (Fig. S1) and potentially played an important role in supplying available P in this meadow steppe (Fig. 6b & Fig. 7). The significant increase of $Ca_{10}$-P with $Ca(H_2PO_4)_2$ addition was not expected; and it was inconsistent with a previous study from a calcareous soil showing unchanged $Ca_{10}$-P after 21-years of superphosphate application (Wang et al.,

2010). In this calcareous soil, $Ca(H_2PO_4)_2$ addition enhanced the transformation of moderate-cycling IP fractions ($Ca_2$-P and $Ca_8$-P) into more stable $Ca_{10}$-P fractions for reasons that are not clear. In return, $Ca_{10}$-P can be a potential P sink of moderate-cycling IP pools because of its significant correlation with Fe-P under both chemical P forms (Table 3). Soil O-P showed a hump-shaped relationship with P

addition for both P compounds when N was also supplied, while no relationship was

found for $KH_2PO_4$ without N addition (Fig. 4c, d). This is in contrast to a 21-year long

study where increased levels of $Ca(H_2PO_4)_2$ significantly increased O-P (Wang et al.,

2010). The discrepancy might be due to 1) differences in soil type affecting soil P

dynamics differently (a Calcarid Regosol in the Wang et al. (2010) study and a

Chernozem in this study); 2) different P compounds having different effects on O-P

cycling; 3) N addition interacting with P to affect plant P uptake and O-P

transformations (Marklein and Houlton, 2012; Zhang et al., 2004). Thus, P addition

effects on O-P and other recalcitrant P fractions were complex, particularly when N

was also added.

The significant increase in soil Olsen-P with P addition levels was consistent with

previous studies (Karaca et al., 2002; Zhou et al., 2018). Significant increase in

Olsen-P most likely was a consequence of inputs of P going into this pool surpassing

output processes of mineral fixation, leaching and plant uptake. Dissolution of

exogenous P compounds could be the primary input process, which then directly

contributed to the increase in Olsen-P concentration. Moreover, the soil IP pools

associated with moderate-cycling of P, especially Fe-P, Al-P, $Ca_2$-P and $Ca_8$-P, can also

be an important source of bioavailable P (Zhang et al., 2012). Significantly positive

correlations between moderate-cycling IP fractions and Olsen-P (Table 3; Fig. 6)

suggest that moderate-cycling P fractions contributed strongly to enhance soil P

availability (Fig. 7). Indeed, moderate-cycling IP fractions could release soil available

P more easily than recalcitrant fractions (Zhang et al., 2012).

### 4.2. Effects of compound-specific P additions

Consistent with our second hypothesis, soil moderate-cycling P fractions of Al-P, Fe-P,

$Ca_2$-P, $Ca_8$-P tended to be higher with soluble $KH_2PO_4$ addition than with less-soluble

$Ca(H_2PO_4)_2$ addition with significant increases in some P addition levels (Fig. 2 & Fig.





3). Compared to $Ca(H_2PO_4)_2$, $KH_2PO_4$ is more effective in elevating soil phosphate levels to form moderate-cycling IP fractions by rapidly interacting with Fe- and Al-oxides and $CaCO_3$ ( Havlin et al., 2005). The negative correlation between soil pH and moderate-cycling IP fractions with $Ca(H_2PO_4)_2$ addition further suggests that the

5 decrease in soil pH contributed to the increase in moderate-cycling IP fractions with $Ca(H_2PO_4)_2$ addition (Fig. 6b). Lower soil pH could accelerate recalcitrant IP fractions converting into moderate-cycling IP and Olsen P (Alt et al., 2013). Under the $Ca(H_2PO_4)_2$ treatment, significant negative relationships of recalcitrant IP and Olsen P further support this argument (Fig. 6b). On the other hand, plant biomass instead of

10 soil pH was more responsible for variations in moderate-cycling IP fractions with $KH_2PO_4$ addition (Fig. 6a). Under both P compounds, moderate-cycling IP fractions contributed to the increase of soil Olsen P across N treatments (Fig. 6), which suggested that higher plant growth and P demand (Fig. 1c,d) could enhance the conversion of moderate-cycling IP fraction into available P.

15  In contrast to our expectation, the recalcitrant IP fractions of O-P and $Ca_{10}$-P were higher with $Ca(H_2PO_4)_2$ than with $KH_2PO_4$ addition (Fig. 4 & Fig. 7). This suggests that $Ca(H_2PO_4)_2$ could be converted more rapidly into stable inorganic P forms than the more soluble $KH_2PO_4$. Additionally, lower O-P and $Ca_{10}$-P concentrations could be caused by enhanced plant P uptake intensity under $KH_2PO_4$ addition which

20 facilitated more recalcitrant IP fractions transforming into moderate-cycling IP and Olsen P. Thus, $Ca(H_2PO_4)_2$ appears to be better than the $KH_2PO_4$ in maintaining recalcitrant IP fractions in the soil. Although soil recalcitrant IP fractions (O-P and $Ca_{10}$-P) are relatively stable, they play essential roles in buffering the depletion of moderate-cycling P and maintaining soil available P levels (Seeling and Jungk, 1996;

25 Vu et al., 2008).



As expected, more soluble $KH_2PO_4$ could result in higher available P (Fig. 7) in
soil solution through dissolution. It is generally observed that this more soluble P
fertilizer type dissolves in soil water at a shorter time and generates more free $PO_4^{3-}$
(Chien et al., 2011). Moderate-cycling IP fractions also contributed to elevated

Olsen-P concentrations with stronger correlations under $KH_2PO_4$ addition (Table 3).
Moderate-cycling IP factions showed a stronger direct and positive effect on Olsen-P
with $KH_2PO_4$ addition than $Ca(H_2PO_4)_2$ addition, while plant biomass showed both
indirect negative (by negatively affecting moderate-cycling IP) and positive effects on
Olsen-P with $KH_2PO_4$ addition (Fig. 6). Based on the correlation analyses between

soil IP fractions and Olsen-P, we found that $Ca_2$-P and Al-P were the dominant IP
fractions to improve soil P availability in the calcareous soil. Our results suggest that
$KH_2PO_4$ was better than $Ca(H_2PO_4)_2$ in alleviating P limitations of soil
microorganisms and plants by promoting the formation of moderate-cycling IP
fractions and Olsen-P in the semi-arid steppe.

**4.3. Nitrogen addition regulated P effects on soil inorganic P fractions**

We predicted that the concentrations of soil IP fractions would be lower under
combined P and N additions than under P addition alone because of increased plant P
uptake with N addition (third hypothesis). We found some support for this, where
combined P and N additions decreased most soil IP fractions compared to P addition

alone, except for the $Ca_{10}$-P fraction. We also found a significant increase in plant
biomass with combined P and N addition (Fig. 1a,b) and a negative correlation
between plant biomass and moderate-cycling IP fractions (Fig. 6a,b). Therefore, the
decrease in soil moderate-cycling IP fractions with N addition could have been due to
enhanced plant P uptake (Fig. 1c,d) as a result of increased plant biomass (Fig. 1a,b &

Fig. 7). Under N addition, simultaneous increases in Olsen-P output (plant uptake)

and input pathways (transformation from moderate-cycling IP fractions) may have resulted in mostly non-significant difference in Olsen-P concentrations between combined P and N addition and P addition alone (Fig. 5a, b). Nitrogen addition can potentially increase soil P availability by promoting solubilization of IP fractions in

the short-term (Wang et al., 2016). However, long-term N deposition resulted in soil IP-exhaustion, thereby constraining the growth of plants (Olander and Vitousek, 2000; Yang et al., 2014). Previous research has also found that decades of N addition could accelerate $PO_4^{3-}$ release (Malik et al., 2012; Stroia et al., 2011) and enhance conversion of recalcitrant IP fractions to labile and moderate-cycling IP fractions as a

result of soil acidification (Alt et al., 2013). Additionally, N addition was found to suppress acid and alkaline phosphatase enzymes resulting in the decrease of soil organic P mineralization in a similar semi-arid grassland (Tian et al. 2016). Therefore, combined N and P addition might decrease soil IP fractions by reducing the conversion of organic P to IP as compared to P addition alone. Our results clearly

illustrate that N effects on soil IP fractions depended on P inputs, where combined N and P additions could accelerate conversion of soil moderate-cycling IP fractions into available P and enhance plant P uptake and biomass (Fig. 7).

### 5. Conclusions

Addition of P compounds significantly increased soil moderate-cycling IP fractions of Al-P, Fe-P, $Ca_2$-P and $Ca_8$-P, which may have contributed to higher available P (Olsen P) in the soil. Soil moderate-cycling IP fractions were higher with soluble $KH_2PO_4$ addition, but in contrast, recalcitrant fractions of $Ca_{10}$-P and O-P were higher with $Ca(H_2PO_4)_2$ addition. Combined N and P addition decreased soil IP fractions due to

enhanced plant P uptake compared to P addition alone for both P compounds. Thus, N





addition promoted the transformation of moderate and recalcitrant IP fractions into available forms. Moderate-cycling IP fractions had a greater contribution to soil P availability than recalcitrant P fractions. Overall, our findings elucidated the interactive effects of N and P addition on soil IP dynamics and presented the first

evidence for the relative roles of exogenous P compounds in regulating P availability in the meadow steppe grassland.

**Acknowledgments**

We acknowledge Erguna Forest-Steppe Ecotone Ecosystem Research Station,

Institute of Applied Ecology, Chinese Academy of Sciences for logistical support. The study was financially supported by the National Natural Science Foundation of China (31770525) and the National Key Research and Development Program of China (2016YFC0500707).

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





**Tables**

**Table 1** General characteristics of the soil from the experimental site before nitrogen and phosphorus manipulation. Selected parameters include sand, silt and clay fractions, soil pH, soil organic matter (SOM), total nitrogen (TN), total phosphorus (TP), total inorganic phosphorus (TIP), organic P, exchangeable Ca and Al, and available Fe and Mn. Data are means ±SE (n = 5).

| Sand (%) | Silt (%) | Clay (%) | Soil pH | SOM (g kg$^{-1}$) | TN (g kg$^{-1}$) | TP (mg kg$^{-1}$) | TIP (mg kg$^{-1}$) | Organic P (mg kg$^{-1}$) | Ca (cmol kg$^{-1}$) | Al (cmol kg$^{-1}$) | Fe (mg kg$^{-1}$) | Mn (mg kg$^{-1}$) |
|---|---|---|---|---|---|---|---|---|---|---|---|---|
| 36.75 ± 0.93 | 39.61 ± 0.96 | 23.64 ± 0.78 | 6.68 ± 0.06 | 43.89 ± 1.14 | 1.83 ± 0.06 | 508.23 ± 21.30 | 105.05 ± 2.87 | 403.17 ± 22.95 | 13.41 ± 0.43 | 0.05 ± 0.01 | 36.76 ± 1.77 | 30.70 ± 0.93 |





**Table 2** Results ($F$ values) of three-way ANOVAs on the effect of phosphorus (P)

fertilizer type ($P_t$), P addition rate ($P_r$), nitrogen (N) addition and their interactions on

soil inorganic phosphorus fractions of variscite (Al-P), strengite (Fe-P), dicalcium

phosphate ($Ca_2$-P), octacalcium phosphate ($Ca_8$-P), hydroxylapatite ($Ca_{10}$-P),

5   occluded phosphate (O-P), Olsen-P, total inorganic phosphorus(TIP), soil total P (TP)

with $KH_2PO_4$ and $Ca(H_2PO_4)_2$ addition.

| | Al-P | Fe-P | $Ca_2$-P | $Ca_8$-P | $Ca_{10}$-P | O-P | Olsen-P | TIP | TP |
|---|---|---|---|---|---|---|---|---|---|
| $P_t$ | $32.4^{**}$ | $76.7^{**}$ | $59.9^{**}$ | $101^{**}$ | $98.1^{**}$ | $0.60$ | $21.5^{**}$ | $60.4^{**}$ | $2.90$ |
| $P_r$ | $268^{**}$ | $97.3^{**}$ | $73.8^{**}$ | $62.6^{**}$ | $6.30^{**}$ | $3.88^{*}$ | $25.0^{**}$ | $192^{**}$ | $7.70^{**}$ |
| N | $37.1^{**}$ | $18.2^{**}$ | $11.1^{**}$ | $21.9^{**}$ | $5.20^{*}$ | $0.00$ | $1.90$ | $36.8^{**}$ | $3.80$ |
| $P_t \times P_r$ | $7.90^{**}$ | $3.40^{*}$ | $11.6^{**}$ | $8.40^{**}$ | $1.00$ | $3.21^{*}$ | $5.90^{**}$ | $13.6^{**}$ | $0.90$ |
| $P_t \times N$ | $0.40$ | $0.10$ | $5.30$ | $6.20$ | $0.10$ | $19.7^{**}$ | $1.80$ | $0.00$ | $6.00^{*}$ |
| $P_r \times N$ | $6.30^{**}$ | $2.10$ | $2.70^{*}$ | $2.20$ | $0.60$ | $0.70$ | $2.00$ | $4.90^{**}$ | $0.80$ |
| $P_t \times P_r \times N$ | $4.20^{**}$ | $2.60^{*}$ | $0.90$ | $4.00^{**}$ | $0.40$ | $2.81^{*}$ | $1.30$ | $3.40^{**}$ | $3.90^{**}$ |

$^{*}$, $^{**}$ Significance level at 0.05 and 0.01, respectively





**Table 3** Correlation analyses (*R* values) among soil inorganic fractions and Olsen-P

with $Ca(H_2PO_4)_2$ and $KH_2PO_4$ addition in the meadow steppe.

| | | Fe-P | $Ca_2$-P | $Ca_8$-P | $Ca_{10}$-P | O-P | Olsen-P |
|---|---|---|---|---|---|---|---|
| $KH_2PO_4$ addition | Al-P | 0.92** | 0.93** | 0.87** | 0.25 | 0.15 | 0.70** |
| | Fe-P | | 0.95** | 0.86** | 0.27* | 0.19 | 0.59** |
| | $Ca_2$-P | | | 0.92** | 0.23 | 0.12 | 0.63** |
| | $Ca_8$-P | | | | 0.29* | 0.10 | 0.57** |
| | $Ca_{10}$-P | | | | | 0.00 | 0.17 |
| | O-P | | | | | | 0.07 |
| $Ca(H_2PO_4)_2$ addition | Al-P | 0.91** | 0.92** | 0.80** | 0.47* | -0.04 | 0.48** |
| | Fe-P | | 0.94** | 0.87** | 0.49* | -0.03 | 0.46** |
| | $Ca_2$-P | | | 0.84** | 0.42 | -0.03 | 0.53** |
| | $Ca_8$-P | | | | 0.50* | 0.05 | 0.34** |
| | $Ca_{10}$-P | | | | | 0.26* | 0.11 |
| | O-P | | | | | | -0.29* |

*, ** Significance level at 0.05 and 0.01, respectively

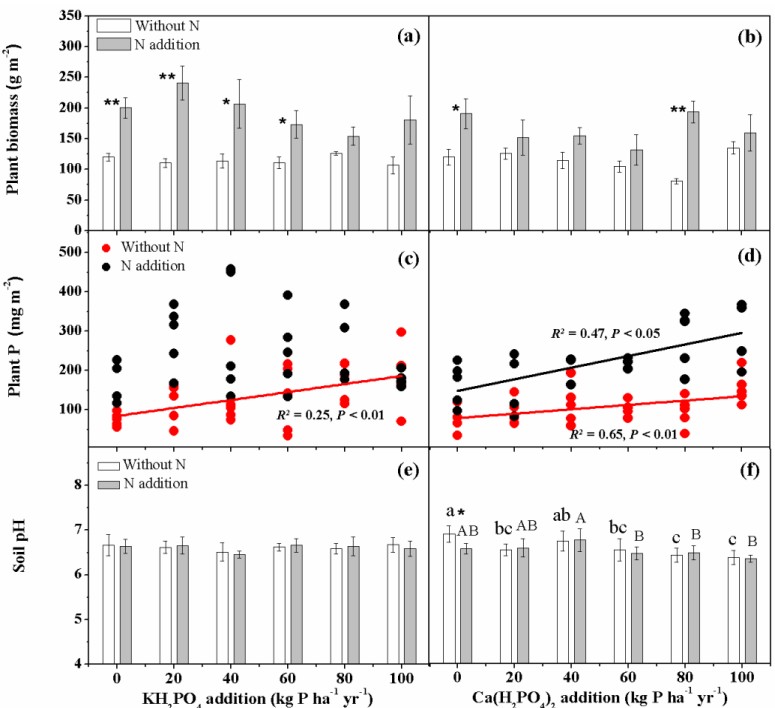

**Fig. 1** Aboveground plant biomass (a, b), biomass-weighted phosphorus (P) uptake of

three dominant plant species (c, d) and soil pH (e, f) as affected by $KH_2PO_4$ and

$Ca(H_2PO_4)_2$ additions with and without N inputs. Data are represented as means $\pm$ SE

5    for panels a, b, e and f. Letters indicate significant differences between P rates of

$KH_2PO_4$ or $Ca(H_2PO_4)_2$ addition without N (lowercase letters) and with N addition

(capital letters). Asterisks represent significance between N treatments within each P

type and rate (* and ** for $P < 0.05$ and 0.01, respectively).



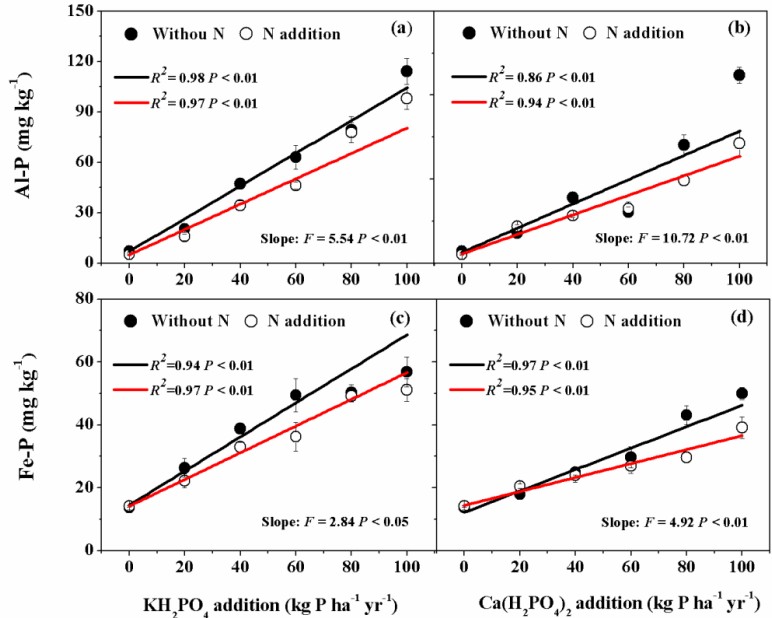

**Fig. 2** Effects of phosphorus (P) and nitrogen (N) additions on soil variscite (Al-P)

and strengite (Fe-P). Data are represented as means ± standard error. Phosphorus

types include $KH_2PO_4$ and $Ca(H_2PO_4)_2$ at rates of 0, 20, 40, 60, 80 and 100 kg P ha$^{-1}$

5   yr$^{-1}$. Fitted lines are based on linear regression models.The black and red line

represent without N and N addition, respectively. Significance was labled for slopes

of the black and red lines.

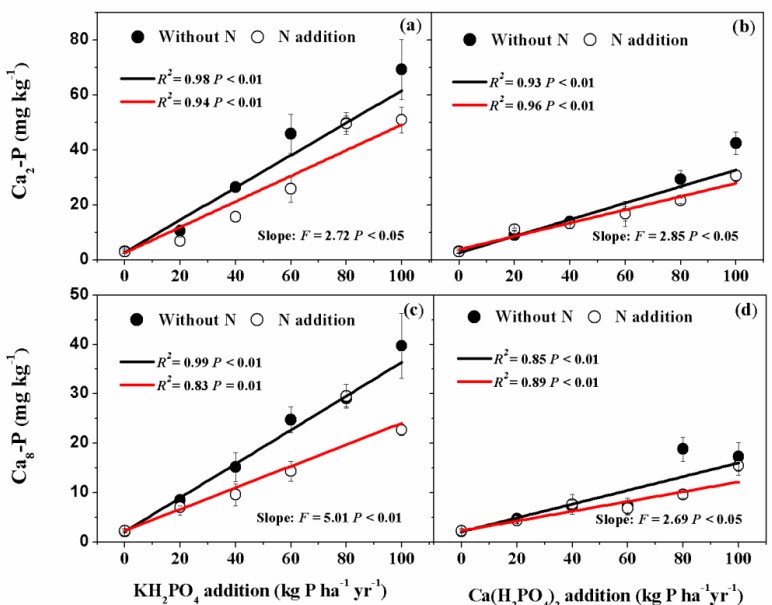

**Fig. 3** Effects of phosphorus (P) and nitrogen (N) additions on soil dicalcium

phosphate ($Ca_2$-P) and octacalcium phosphate ($Ca_8$-P). Phosphorus types include

$KH_2PO_4$ and $Ca(H_2PO_4)_2$ at rates of 0, 20, 40, 60, 80 and 100 kg P ha$^{-1}$ yr$^{-1}$. Data are

5     represented as means ± standard error. Fitted lines are based onlinear regression

models.The black and red line represent without N and N addition, respectively.

Significance was labled for slopes of the black and red lines.


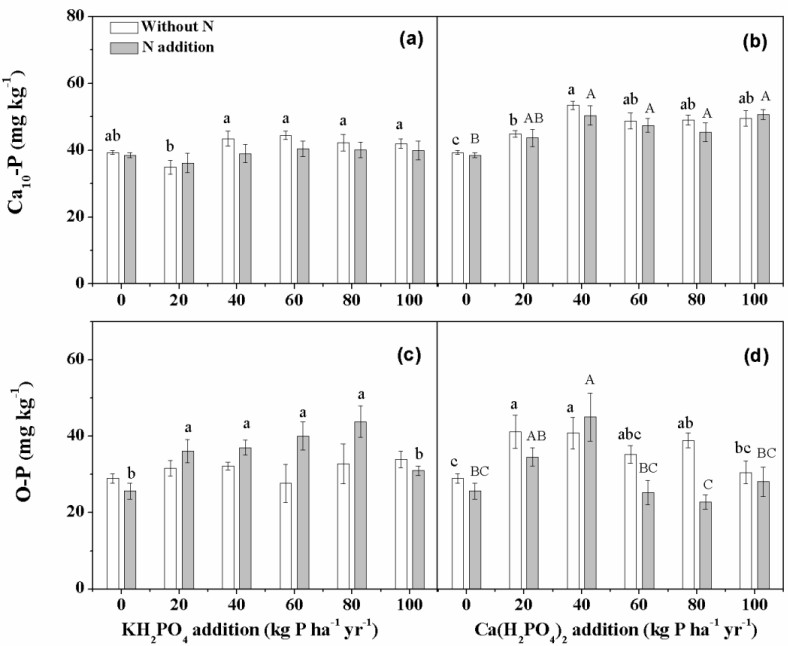

**Fig. 4** Effects of phosphorus (P) and nitrogen (N) additions on soil hydroxylapatite
($Ca_{10}$-P) and occluded P (O-P). Phosphorus types include $KH_2PO_4$ and $Ca(H_2PO_4)_2$ at
rates of 0, 20, 40, 60, 80 and 100 kg P ha$^{-1}$ yr$^{-1}$. Data are represented as means $\pm$
standard error. Letters indicate significant differences between P rates of $KH_2PO_4$ or
$Ca(H_2PO_4)_2$ addition without N (lowercase letters) and with N addition (capital
letters). Asterisks represent significance between N treatments within each P type and
rate (* and ** for P < 0.05 and 0.01, respectively).



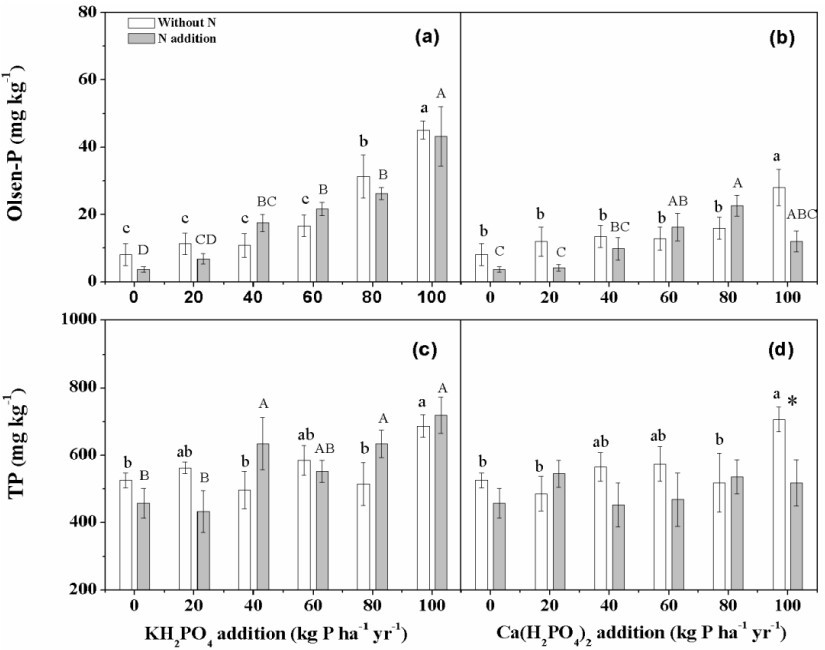

**Fig. 5** Effects of phosphorus (P) and nitrogen (N) additions on soil Olsen-P and total P

(TP) in the meadow steppe. Phosphorus types include $KH_2PO_4$ and $Ca(H_2PO_4)_2$ at

rates of 0, 20, 40, 60, 80 and 100 kg P ha$^{-1}$ yr$^{-1}$. Data are represented as

5      means±standard error. Letters indicate significant differences between P rates of

$KH_2PO_4$ or $Ca(H_2PO_4)_2$ addition without N (lowercase letters) and with N addition

(capital letters). Asterisks represent significance between N treatments within each P

type and rate (* and ** for $P < 0.05$ and 0.01, respectively).



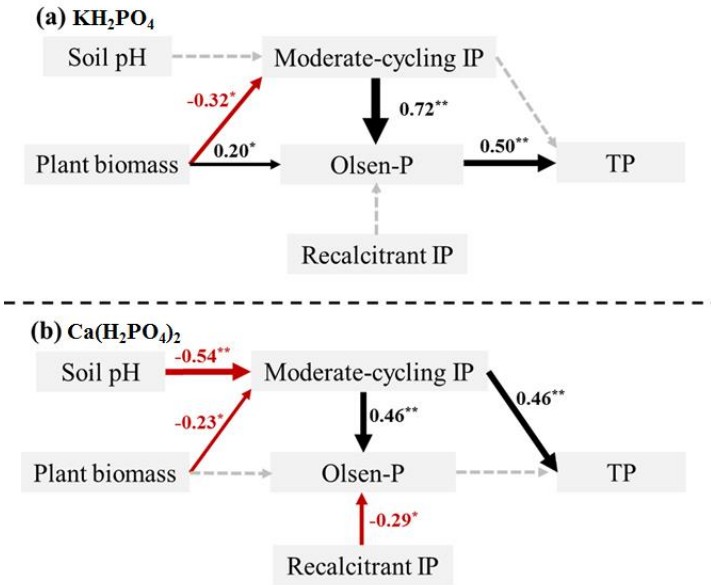

**Fig. 6** Structural equation model of the relationships between soil pH, aboveground

plant biomass, moderate-cycling inorganic phosphorus (IP), recalcitrant IP, Olsen-P

and TP under $KH_2PO_4$ (a, $\chi^2 = 9.37$, $P = 0.59$, RMSEA $= 0.00$, AIC $= 41.37$) and

5     $Ca(H_2PO_4)_2$ additions (b, $\chi^2 = 15.14$, $P = 0.13$, RMSEA $= 0.09$, AIC $= 49.14$) across

N treatments. Arrows indicate positive (black), negative (red) and neutral (grey-dotted)

effects. The number adjacent to each arrow is the standardized path coefficient with

corresponding significance (*, ** for $P < 0.05$ and 0.01, respectively).





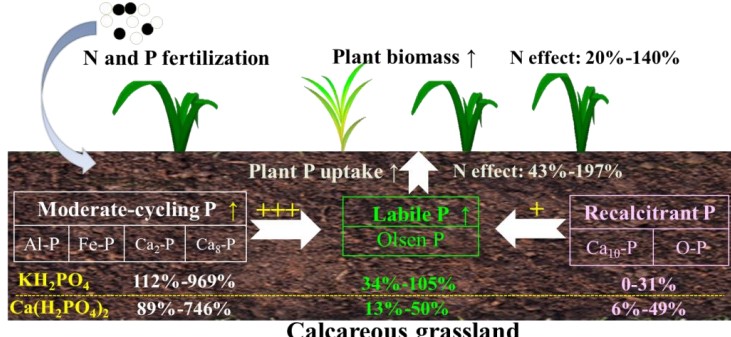

Calcareous grassland

**Fig. 7** Schematic diagram shows the effects of phosphorus (P) compound addition on soil inorganic P transformations in the calcareous grassland of Inner Mongolia. The percentages represent increase ("↑" in the diagram) of soil inorganic P fractions, plant

5   P uptake and plant biomass as affected by fertilization across P addition rates. $KH_2PO_4$ addition had a larger impact on moderate-cycling P (sum of Al-P, Fe-P, $Ca_2$-P and $Ca_8$-P) as compared to $Ca(H_2PO_4)_2$ (112%-969% *vs.* 89%-746%). However, recalcitrant P (sum of $Ca_{10}$-P and O-P) increased more with $Ca(H_2PO_4)_2$ addition relative to $KH_2PO_4$ (6%-49% *vs.* 0-31%). Nitrogen addition decreased

10   moderate-cycling P by enhancing plant biomass and plant P uptake for both P compound additions across P addition rates. Therefore, moderate-cycling P showed a higher contribution to soil labile P than recalcitrant P as represented by '+++' and '+', respectively.