# Peer review of "Table S1 The results of ANCOVA (*F* values) testing the differences in slope between the two N treatments (N effect, without N vs. with N) for each P compound and between the two P compounds (P compound effect, K2HPO4 vs. Ca(H2PO4)2) for eac"

_Biogeosciences, 2019_

## Referee Comment (RC1) · Anonymous Referee #1 · 6 Jul 2019

General comments

This study examined the effects of P compounds ($KH_2PO_4$ and $Ca(H_2O_4)_2$) with different addition rate on soil inorganic P (IP) fraction in a calcareous grassland. This study also set with and without N treatment and compare the effects of N addition on soil IP fraction. The design of this field fertilization experiments is appropreate and some results are fascinating. This study is also relevant to the scope of this journal. I thus think this study is novel and interesting and promote understanding about the phosphorus cycling in dry area. However, I have several concerns on this study and main 3 points are listed below. This paper can be accepted only after revision of this paper

considering the comments. Especially for third points, authors must re-read through the manuscript and revise the relevant table, figure and description.

1. I wonder whether authors could extract Ca2-P and Olsen-P properly. I agree that the potential limitation of the sequential extraction methods as described in p. 8. line 18-20 but I think that authors have to refer to the difference between Ca2-P and Olsen-P in Introduction or Materials and methods. Although both of them are extracted with NaHCO3, Ca2-P is classified into moderate-cycling IP and Olsen-P into labile-P. Moreover, the concentration of Ca2-P and Olsen-P in each treatment are really similar (Fig. 3 a,b and Fig. 5 a,b) and they are significantly positively correlated each other (r= 0.63 and 0.53 in Table 3). I wonder if authors extracted almost same chemical properties. Refer difference between Ca2-P and Olsen-P in terms of chemical compounds. If they are chemically similar compounds, Ca2-P should be eliminated from Moderate-cycling IP when SEM were built.

2. I have some concern about SEM. At first, how do the authors handle N addition effects when making SEM? There is no description about that. Second, the description "Moderate-cycling IP was mainly regulated by plant biomass" (e.g. p.2 line 14-15) sounds strange. In this study, plant biomass is also affected by soil IP. Finally, not only IP but Organic P also affects TP because the percentage of each IP fraction to total P (TP) is less than 50%. Organic P (TP-TIP) fraction accounts large part of TP and must have great effects on TP in this study. Author should explain these three points and reanalyze them.

3. I have serious concern about description in Results section of this study. This paper has several mistakes in the Results section. Some description in Results do not correspond to the relevant Table and Figure. Some specific comments are written below.

Specific comments

p.4 line 16-18: It may be appropriate to show which fraction of inorganic P increased in

the previous study.

p. 6 line 4-7 (2nd hypothesis): I cannot understand how hypothesis 2 was derived. I particularly do not understand why authors hypothesized that recalcitrant P would be higher with KH2PO4 addition than with Ca(H2PO4)2 addition. I think that it takes much time to form recalcitrant P from labile-P. Authors should add explanation about chemical properties of recalcitrant IP with introducing previous studies.

p.4 line 16-20: These two sentences introduce the previous studies which results were opposite. The connection of them is not good. I think that these two sentences should change to "Some previous study showed that∼∼∼ and other one showed that ∼∼∼. "

It may be appropriate to explain how large three species of plants (Stipa baicalensis, Leymus chinensis and Carex duriuscula) are occupied per area if authors have some data. Is there difference of dominant ratio among treatments?

p. 9 line 3-5 and p.10 line 11 and other: "P uptake" should be replaced by another word (i.e. the biomass-weighted P concentration). In this study, authors just determined the concentration of P in plant body not the P uptake.

Explain about dataset authors have. According to Materials and methods, authors used 110 plots in total including replication, which means that 0 kg P ha-1 yr-1 treatments are overlapped between KH2PO4 and Ca(H2O4)2 fertilization. Although most of their data looks same among KH2PO4 and Ca(H2O4)2 fertilization when 0 kg P were added but some are different. For example, soil pH is almost same between N treatment when 0 kg KH2PO4 were added (Fig. 1e) but there is difference between N treatment when 0 kg Ca(H2O4)2 were added. Explain why some data are difference when 0 kg P were added as seen in soil pH.

p.10 line 21-23: Unclear, explain more in detail

Results 3.3 and 3.4: There are many faults. Relevant Figure and Table does not show the result which are written in manuscript. First of all, it is strange that there are two

"3.3" section. Specific comments are below.

p.11 line 10-13: Fig 4a, b and Table S2 does not show the results as described in manuscript. I think authors should replace "0, 40, 100" by "20, 60, 80" according to Table S2. Moreover, Table S2 just shows whether there is difference of soil IP fraction between types of added P compounds ($KH_2PO_4$ and $Ca(H_2O_4)_2$) or not but it does not show which is higher or lower between them.

However, Table S2 does not coincide with Fig.4a and b. For example, Table S2 shows that soil Ca10-P are significantly different (Student t-test, $p<0.01$) between $KH_2PO_4$ and $Ca(H_2PO_4)_2$ addition when 0 kg P were added but they look no different in Fig 4a and b. I think something is wrong and authors should reconfirm the dataset and reanalyze them.

p.11 line 14-16: There is no asterisk in Fig. 4d. I cannot judge whether there is significant difference between Without N and N addition at 60 and 80 kg P ha-1 yr-1.

p.11 line18-20: Unclear, explain more in detail. What about O-P for 0 kg P ha-1 yr-1 treatment with N addition? Table S2 shows the significant difference between $KH_2PO_4$ and $Ca(H_2PO_4)_2$ addition ($p<0.05$).

p.12 line 3-4: Fig. 5d does not show that results. Soil TP with $Ca(H_2PO_4)_2$ does not increase with increasing P addition when N was added.

p.12 line 7-8: There are no results which indicate correlation between TP and moderate-inorganic IP (Fe-P, Al-P, Ca2-P, Ca8-P) in Table 3.

p.13 line 1: "applied P was immobilized mainly into inorganic forms" sounds strange. "Immobilization" is the conversion of inorganic materials to organic ones.

Add "soil" before each P fraction. Some are already added but I think that authors should unify the expression about soil IP fraction.

Is the term "recalcitrant inorganic phosphorus" used commonly?

p.15 line 13: not "P demand" but "concentration of P in plant" as pointed out above.

p. 15 line 3-6: The sentence is wordy.

Conclusion: I think that authors should suggest how to fertilize P and N appropriately on grassland ecosystems to maintain plant productivity in Conclusion based on the results of this study.

Technical corrections

p.3 line 17 and other: Olsen P => Olsen-P

p.4 line17: inorganic P => IP

p.7 line 1: phosphorus => P

Caption of Fig. 7: moderate-cycling P => moderate-cycling IP

Fig. 4c: Replace lowercase letters (i.e. a, b) by capital letters (i.e. A, B).

Fig. 5a: The vertical axis labels of Fig. 3a and c and the horizontal axis labels of Fig. 5a are written in bold type and they should be changed.

Fig. 6: Explain what the width of arrows means.

Fig. S1 and 2: Replace "N0" and "N10" by "Without N" and "N addition." The terms "N0" and "N10" are not used in the main manuscript.

I recommend that authors use consistent color for "Without N" and "N addition" in figures for easy understand. In current manuscript, white bar represents "Without N" in the bar graph, whereas white dot represents "N addition" in the scatter plots. Red fitted line represents "Without N" in Fig. 1c, d and "N addition" in Fig. 2, 3.

―――――――――――――――――

---

## Referee Comment (RC2) · Anonymous Referee #2 · 16 Jul 2019

Nutrient management in intensively utilized grasslands is an essential topic that draws considerable attention in current studies. This study of P and N manipulation was done in an important ecosystem where a large population is supported. It has the potential to address the key question on soil P transformation and to advance our understanding of how P and N interacting to affect soil fertility of the meadow grasslands. However, there some minor issues should be properly addressed before being acceptable for publication in Biogeosciences. P2 Line 14-15: logic leaps exist in the statement 'Moderate-cycling IP was mainly regulated by aboveground plant biomass with KH2PO4 addition, while by soil pH and plant biomass with addition of Ca(H2PO4)2'. Please change 'plant biomass' into 'plant P uptake'. P2 Line 24: Please change the

keyword 'soil phosphorus' into 'nitrogen enrichment'. P3 Line 3: change '50-90%' into '50%-90%'. P3 Line 17-18: 'labile/ available P (Olsen P), moderate-cycling P fractions (Al-P, Fe-P, Ca2-P and Ca8-P)'. according to the Methods introduced by the authors, 'Olsen-P was extracted from air-dried soil with 0.5 M NaHCO3 (pH 8.5)(Olsen et al., 1954)', while 'Ca2-P was determined by shaking 0.5 g soil with 25 ml 0.25 M NaHCO3 (pH 7.5)' , and in contrast to Fig. 3a,b and Fig5.a,b, Ca2-P is more suitable to classified into 'labile' fraction. The statement should be 'labile P (Ca2-P)/ available P (Olsen P), moderate-cycling P fractions (Al-P, Fe-P, and Ca8-P)'. P3 Line 21: delete 'understanding'. P4 Line 1: reword 'content' into 'concentration'. P5 Line 6: suggest to rephrase 'release of P from soil organic P' into 'mineralization of soil organic P'. P5 Line 25: 'soil labile IP (Olsen-P)': 'Soil labile IP (Ca2-P) or available P (Olsen-P)'. P6 Line 1: remove 'Ca2-P'. P7 Line 12-13: maybe, this sentence should be written as 'The plants were sorted to species and oven-dried at 65 for 48 h, then weighed and ground'? Because the way you did these should be to determined aboveground net primary productivity and get plant samples ground to measure plant P concentration. P7 Line 24: change 0.5g into 0.5 g. P8 Line 16-18: change 'Total IP (TIP) concentration was defined as the sum of moderate-cycling IP (Al-P, Fe-P, Ca2-P and Ca8-P) and recalcitrant IP (Ca10-P and O-P).' into 'Total IP (TIP) concentration was defined as the sum of liable IP (Ca2-P), moderate-cycling IP (Al-P, Fe-P, and Ca8-P) and recalcitrant IP (Ca10-P and O-P).' P9 Line 24: plant P uptake should be incorporated in the SEM model instead of plant biomass. If plant biomass is included, it should be explained by P fractions/ P availability. But do not use plant biomass to explain P fractions. P 12 Line 12: Even though plant biomass production consume soil P, it would be better to directly use the parameter of plant P uptake as you calculated in P 9 Line 5. P15 Line 16-17: In this study, you were unable to determine the relative transformation rates of the two compounds. But the results of higher O-P and Ca10-P suggested that a higher proportion of Ca(H2PO4)2 was converted into stable inorganic P forms. So, please correct this point. P16 Line 7 & 22: Be aware that the mechanistic description might change if you replace the parameter of plant biomass with plant P uptake. P18 Line 19-20 change

'Phosphate Release Kinetics in Calcareous Grassland and Forest Soils in Response to H+ Addition' into 'Phosphate release kinetics in calcareous grassland and forest soils in response to H+ addition' P19 Line 9: Lolium perenne and Pinus radiate, italic. P20 Line 1-2: change 'Do Nutrient Limitation Patterns Shift from Nitrogen Toward Phosphorus with Increasing Nitrogen Deposition Across the Northeastern United States?' into 'Do nutrient limitation patterns shift from nitrogen toward phosphorus with increasing nitrogen deposition across the Northeastern United States?' P20 Line 24-25: change 'Soil fertility and fertilizers an introduction to nutrient management' into 'Soil Fertility and Fertilizers: An Introduction to Nutrient Management' P24 Line 20-21: change 'Nitrogen Fertilization Effects on Grassland Soil Acidification: Consequences on Diffusive Phosphorus Ions' into 'Nitrogen fertilization effects on grassland soil acidification: consequences on diffusive phosphorus ions' P26 Line 1: 'Larix gmelinii', italic? P30 Line 8: 'P', italic. P33 Line 8: 'P', italic. P35 Fig. 6: reshape the 'Structural equation model' and rewrite the Results and Discussion sections related to the SEM. P36 Fig. 7: The content in the figure is hard to read because of the background color. Remove Ca2-P from the 'moderate-cycling P', and instead of using Olsen as labile P, I suggest the authors use Ca2-P as labile P.

---

## Author Comment (AC1) · 18 Aug 2019

We sincerely thank the reviewer for the constructive comments and suggestions, which helped us to substantially improve our manuscript. Please find the point-to-point responses (blue) to the comments (black) as listed below.

**Reviewer 1**

**Comment 1:** I wonder whether authors could extract $Ca_2$-P and Olsen-P properly. I agree that the potential limitation of the sequential extraction methods as described in p. 8. Line 18-20 but I think that authors have to refer to the difference between $Ca_2$-P and Olsen-P in Introduction or Materials and methods. Although both of them are extracted with $NaHCO_3$, $Ca_2$-P is classified into moderate-cycling IP and Olsen-P into labile-P. Moreover, the concentration of $Ca_2$-P and Olsen-P in each treatment are really similar (Fig.3 a,b and Fig. 5 a,b) and they are significantly positively correlated each other (r= 0.63 and 0.53 in Table 3). I wonder if authors extracted almost same chemical properties. Refer difference between $Ca_2$-P and Olsen-P in terms of chemical compounds. If they are chemically similar compounds, $Ca_2$-P should be eliminated from Moderate-cycling IP when SEM were built.

**Response:** Thanks for the comment. And we agree with the reviewer's comment that $Ca_2$-P and Olsen P are chemically similar. The description about the difference between $Ca_2$-P and Olsen-P was added in the Introduction section as "Soil Olsen P could be directly absorbed and utilized by plants, which includes all water-soluble P, some of the absorbed and soluble IP, and mineralizable organic P (Tang et al., 2009; Cao et al., 2012). $Ca_2$-P, chemically similar to Olsen P, includes water-soluble P, citrate-soluble P, and partially surface-adsorbed P (Shen et al., 2004; Zhao et al., 2019)" (P. 4 Lines 23-24). Therefore, we defined $Ca_2$-P as labile IP and Olsen-P as available P (P. 4 Lines 16-19). And the SEM has been reanalyzed by eliminating $Ca_2$-P from moderate-cycling IP, where it was defined as the labile IP in the SEM modeling.

*Reference cited:*

*Shen, J., Li, R., Zhang, F., Fan, J., Tang, C., Renfenl, Z.: Crop yields, soil fertility and phosphorus fractions in response to long-term fertilization under the rice monoculture system on a calcareous soil, Filed Crop Res. 86, 225-238.*

*Tang, X., Ma, Y., Hao, X., Li, X., Li, J., Huang, S., Yang, X.: Determining critical*

*values of soil Olsen-P for maize and winter wheat from long-term experiments in China, Plant Soil, 323:143-151.*

*Cao, N., Chen, X., Cui, Z., Zhang, F.: Change in soil available phosphorus in relation to the phosphorus budget in China, Nutr. Cycl. Agroecosyst, 94: 161-170, 10.1007/s10705-012-9530-0, 2012.*

*Zhao, F., Zhang, Y., Dijkstra, F. A., Li, Z., Zhang, Y., Zhang, T., Lu, Y., Shi, J., and Yang, L.: Effect of amendments on phosphorous status in soils with different phosphorous levels, Catena 172, 97-103.*

**Comment 2:** I have some concern about SEM. At first, how do the authors handle N addition effects when making SEM? There is no description about that. Second, the description "Moderate-cycling IP was mainly regulated by plant biomass" (e.g. p.2 line 14-15) sounds strange. In this study, plant biomass is also affected by soil IP. Finally, not only IP but Organic P also affects TP because the percentage of each IP fraction to total P (TP) is less than 50%. Organic P (TP-TIP) fraction accounts large part of TP and must have great effects on TP in this study. Author should explain these three points and reanalyze them.

**Response:** Thanks for the constructive suggestions. We revised the manuscript according to all the suggestions as mentioned above. First, N and P addition effects have been incorporated into the SEM (please see Fig. 6), which was described in the Statistical analysis section as "Structural equation models (SEM) were built to clarify direct and indirect N and P addition effects on soil P fractions through the changes in plant P uptake and soil pH" (P. 11 Lines 7-8). Second, we replaced the parameter of plant biomass with plant P uptake in the SEM, because the plant P uptake intensity (amount) instead of plant biomass can directly affect soil P dynamics. Third, we fully agree with the point that soil organic P has great effects on TP and we tried to add the soil organic P into the SEM. However, there are multiple collinearities (variance proportions = 0.97) between soil organic P and soil TP resulting in the failure of the SEM. Thus, we added the correlation between organic P and TP to illustrate the significant contribution of organic P to TP in Table 3.

**Comment 3:** I have serious concern about description in Results section of this study. This paper has several mistakes in the Results section. Some description in Results do not correspond to the relevant Table and Figure. Some specific comments are written

below.

**Response:** We have addressed these concerns in the following responses to specific comments.

**Specific comments**

p.4 line 16-18: It may be appropriate to show which fraction of inorganic P increased in the previous study.

**Response:** Thanks for the comment. We have added the information of which inorganic P fractions increased in the previous study as "In a previous study it was found that long-term application of $Ca(H_2PO_4)_2$ increased soil IP fractions (Al-P, Fe-P, $Ca_2$-P, $Ca_8$-P and O-P) and total P (TP) concentrations in a calcareous soil (Wang et al. 2010)" (P. 5 Lines 22-24).

p. 6 line 4-7 (2nd hypothesis): I cannot understand how hypothesis 2 was derived. I particularly do not understand why authors hypothesized that recalcitrant P would be higher with $KH_2PO_4$ addition than with $Ca(H_2PO_4)_2$ addition. I think that it takes much time to form recalcitrant P from labile-P. Authors should add explanation about chemical properties of recalcitrant IP with introducing previous studies.

**Response:** Thanks for the comment. We hypothesized that recalcitrant P would be higher with $KH_2PO_4$ addition than with $Ca(H_2PO_4)_2$ addition, because more soluble $KH_2PO_4$ can convert into labile and moderate-cycling IP fractions faster and consequentially promoting the formation of recalcitrant IP fractions. We agree that it takes much time to form recalcitrant P from labile P, but the time should be shorter for $KH_2PO_4$ addition than $Ca(H_2PO_4)_2$. Here, hypothesis 2 has been clarified as "addition of soluble $KH_2PO_4$ would be more efficient in increasing soil labile IP, moderate-cycling IP fractions, recalcitrant P fractions and total P (TP) than less-soluble $Ca(H_2PO_4)_2$, because of faster conversion of $KH_2PO_4$ into labile and moderate-cycling IP fractions and consequentially promoting the formation of recalcitrant fractions" (P. 7 Lines 9-14). We have added chemical properties of soil recalcitrant IP in the introduction section as "Soil recalcitrant IP is relatively stable and unavailable for plants, which is mainly converted from the fixation of labile and moderate-cycling IP (Shen et al., 2004; Zhao et al., 2019)" (P. 4 Line 25-P. 5 Line 1).

We hope this information would help to support the 2nd hypothesis.

*Reference cited:*

*Shen, J., Li, R., Zhang, F., Fan, J., Tang, C., Renfenl, Z.: Crop yields, soil fertility and phosphorus fractions in response to long-term fertilization under the rice monoculture system on a calcareous soil, Filed Crop Res. 86, 225-238, 10.1016/j.fcr.2003.08.013, 2004.*

*Zhao, F., Zhang, Y., Dijkstra, F. A., Li, Z., Zhang, Y., Zhang, T., Lu, Y., Shi, J., and Yang, L.: Effect of amendments on phosphorous atatus in soils with different phosphorous levels,Catena, 172, 97-103, https://doi.org/10.1016/j.catena.2018.08.016, 2019.*

p.4 line 16-20: These two sentences introduce the previous studies which results were opposite. The connection of them is not good. I think that these two sentences should change to "Some previous study showed that~~. "

**Response:** Thanks for the helpful suggestion. This sentence has been corrected accordingly (P. 5 Line 22- P. 6 Line 1).

It may be appropriate to explain how large three species of plants (*Stipa baicalensis, Leymus chinensis and Carex duriuscula*) are occupied per area if authors have some data. Is there difference of dominant ratio among treatments?

**Response:** As suggested, we calculated and analyzed the relative biomass proportion of the three species (shown in P. 8 Lines 2-6) which showed no difference among P addition rates but significantly increased with N addition. The data for relative proportion of the three dominant species has been listed in the following table.

**Table** Sum of the relative biomass (%) of three dominant plant species (*Stipa baicalensis*, *Leymus chinensis* and *Carex duriuscula*) as affected by P addition type and rate without and with N addition, respectively.

| P rate (kg P ha$^{-1}$ yr$^{-1}$) | $KH_2PO_4$ | | $Ca(H_2PO_4)_2$ | |
|---|---|---|---|---|
| | Without N | With N | Without N | With N |
| 0 | 61.07±6.75 | 82.16±4.31 | 61.07±6.75 | 82.16±4.31 |
| 20 | 65.35±8.34 | 80.73±5.62 | 54.56±4.77 | 81.21±4.62 |
| 40 | 63.94±6.83 | 83.15±7.56 | 67.16±6.61 | 85.06±3.96 |
| 60 | 73.61±2.77 | 83.37±6.00 | 50.29±7.12 | 89.05±3.25 |
| 80 | 67.39±6.89 | 89.23±2.30 | 59.60±6.28 | 91.92±2.22 |
| 100 | 74.75±5.86 | 72.43±9.92 | 65.07±3.30 | 83.11±5.30 |

p.9 line 3-5 and p.10 line 11 and other: "P uptake" should be replaced by another word (i.e. the biomass-weighted P concentration). In this study, authors just determined the concentration of P in plant body not the P uptake.

**Response:** We thank the reviewer for the suggestion. But here we calculated the amount of P uptaken by the three dominant species via multiplying P concentration by plant biomass (P. 10 Lines 13-15). It's not biomass-weighted P concentration which is determined via multiplying P concentration by plant biomass proportion.

Explain about dataset authors have. According to Materials and methods, authors used 110 plots in total including replication, which means that 0 kg P ha$^{-1}$ yr$^{-1}$ treatments are overlapped between $KH_2PO_4$ and $Ca(H_2O_4)_2$ fertilization. Although most of their data looks same among $KH_2PO_4$ and $Ca(H_2O_4)_2$ fertilization when 0 kg P were added but some are different. For example, soil pH is almost same between N treatment when 0 kg $KH_2PO_4$ were added (Fig. 1e) but there is difference between N treatment when 0 kg $Ca(H_2O_4)_2$ were added. Explain why some data are difference when 0 kg P were added as seen in soil pH.

**Response:** Sorry for the confusion. In the field experiment, we have 5 replicate control plots for both $KH_2PO_4$ and $Ca(H_2O_4)_2$ fertilization. In our previous version, we forgot to overlap the control plots between $KH_2PO_4$ and $Ca(H_2O_4)_2$ fertilization for soil pH. But it is correct for all the other parameters. Now, we have corrected and reanalyzed the data of soil pH (Fig. 1e, f).

p.10 line 21-23: Unclear, explain more in detail

**Response:** Thanks for the comments. This sentence mainly explained the interactive $P_r \times N$ and $P_t \times P_r \times N$ effects on soil Al-P. It has been clarified into "For instance, Al-P concentration was higher with $KH_2PO_4$ addition than $Ca(H_2PO_4)_2$ at P addition level of 60 kg P ha$^{-1}$ yr$^{-1}$ when N was not added, but it was higher for P addition levels of 60, 80 and 100 kg P ha$^{-1}$ yr$^{-1}$ when N was added" (P. 12 Lines 5-8).

Results 3.3 and 3.4: There are many faults. Relevant Figure and Table does not show the result which are written in manuscript. First of all, it is strange that there are two

"3.3" section. Specific comments are below.

**Response:** Sorry for the confusion. We have changed the second 3.3 to 3.4 and changed 3.4 to 3.5. Responses for all the specific comments are listed below.

p.11 line 10-13: Fig 4a, b and Table S2 does not show the results as described in manuscript. I think authors should replace "0, 40, 100" by "20, 60, 80" according to Table S2. Moreover, Table S2 just shows whether there is difference of soil IP fraction between types of added P compounds ($KH_2PO_4$ and $Ca(H_2O_4)_2$) or not but it does not show which is higher or lower between them. However, Table S2 does not coincide with Fig.4a and b. For example, Table S2 shows that soil $Ca_{10}$-P are significantly different (Student t-test, $p<0.01$) between $KH_2PO_4$ and $Ca(H_2PO_4)_2$ addition when 0 kg P were added but they look no different in Fig 4a and b. I think something is wrong and authors should reconfirm the dataset and reanalyze them.

**Response:** We thank the reviewer for pointing this out. We checked all the data and presented Table S2 with means ± standard error in order to show which P type was higher. According to Table S2, we replaced 0, 40 ,100 by 40, 60, 100. We feel really sorry that we mistakenly did not present the data from the same control of $KH_2PO_4$ addition as $Ca(H_2PO_4)_2$ in Table S2 in our previous version (but the data of 0 kg P presented in Figure 4 were correct) (same correction mentioned for soil pH data). This has been corrected in the current version where we now used the data from the same control plots.

p.11 line 14-16: There is no asterisk in Fig. 4d. I cannot judge whether there is significant difference between Without N and N addition at 60 and 80 kg P ha$^{-1}$ yr$^{-1}$.

**Response:** Sorry for the confusion. We added the asterisks in Fig. 4d.

p.11 line18-20: Unclear, explain more in detail. What about O-P for 0 kg P ha$^{-1}$ yr$^{-1}$ treatment with N addition? Table S2 shows the significant difference between $KH_2PO_4$ and $Ca(H_2PO_4)_2$ addition ($p<0.05$).

**Response:** We thank the reviewer for pointing this out. We feel sorry for our mistake in presenting the data from control plots. The O-P concentration should be the same in the control plots for $KH_2PO_4$ and $Ca(H_2PO_4)_2$ addition, as we used the same control

plots for the two P types as described in the Material and Method section. This has been corrected.

p.12 line 3-4: Fig. 5d does not show that results. Soil TP with $Ca(H_2PO_4)_2$ does not increase with increasing P addition when N was added.

**Response:** Thanks for the observation. This sentence has been corrected into "$KH_2PO_4$ addition increased soil TP irrespective of N addition, while $Ca(H_2PO_4)_2$ addition only increased soil TP without N addition".

p.12 line 7-8: There are no results which indicate correlation between TP and moderate-inorganic IP (Fe-P, Al-P, $Ca_2$-P, $Ca_8$-P) in Table 3.

**Response:** Thanks so much for the observation. We added the correlation between soil TP and moderate-cycling IP (Al-P, Fe-P and $Ca_8$-P) in Table 3.

p.13 line 1: "applied P was immobilized mainly into inorganic forms" sounds strange. "Immobilization" is the conversion of inorganic materials to organic ones.

**Response:** As per suggestion, we replaced the word "immobilized" with "fixed" (P. 14 Line 15).

Add "soil" before each P fraction. Some are already added but I think that authors should unify the expression about soil IP fraction.

**Response:** We agree with the reviewer and have added "soil" before each P fractions in the manuscript.

Is the term "recalcitrant inorganic phosphorus" used commonly?

**Response:** Yes, we found some references about recalcitrant IP. In this study, we defined $Ca_{10}$-P and O-P as the recalcitrant IP which is insoluble and unavailable for plants. Miller et al (2001) introduced that most soil P exists in recalcitrant minerals and forms soil recalcitrant phosphate with the depletion of primary minerals. In addition, Lawrence et al (2001) also proposed the conversion of recalcitrant inorganic phosphorus into other P forms.

*Reference cited:*

*Miller, A., Schuur, E., Chadwick, O.: Redox control of phosphorus pools in Hawaiian*

*montane forest soils, Geoderma, 102: 0-237, 10.1016/s0016-7061(01)00016-7, 2001.*

*Lawrence, D and Schlesinger, W.: Changes in soil phosphorus during 200 years of shifting cultivation in Indonesia, Ecology, 82: 2769-2780, 10.2307/2679959, 2001.*

p.15 line 13: not "P demand" but "concentration of P in plant" as pointed out above.

**Response:** It's not "concentration of P in plant", because plant P uptake (g P per $m^2$) was calculated via multiplying plant P concentration (g P $kg^{-1}$ biomass) by plant biomass (g $m^{-2}$).

p. 15 line 3-6: The sentence is wordy.

**Response:** Thanks for the comments. We reanalyzed the SEM and clarified the description into "The decrease in soil pH contributed to the increase in labile P with $Ca(H_2PO_4)_2$ addition (Fig. 6b)" (P. 16 Lines 20-21) according to the SEM model.

Conclusion: I think that authors should suggest how to fertilize P and N appropriately on grassland ecosystems to maintain plant productivity in Conclusion based on the results of this study.

**Response:** Thanks for the constructive suggestion. This information has been added in the Conclusion section as "Overall, P fertilization is necessary for promoting productivity and sustainable management of grasslands by maintaining soil P availability and pools under scenarios of ecosystem N enrichment" (P. 19 Lines 23-25).

Technical corrections

p.3 line 17 and other: Olsen P => Olsen-P

**Response:** As per suggestion, we replaced "Olsen P" with "Olsen-P".

p.4 line17: inorganic P => IP

**Response:** As per suggestion, we replaced "inorganic P" with "IP".

p.7 line 1: phosphorus => P

**Response:** As per suggestion, we replaced "phosphorus" with "P".

Caption of Fig. 7: moderate-cycling P => moderate-cycling IP

**Response:** As per suggestion, we replaced "moderate-cycling P" with "moderate-cycling IP" in Fig. 7 caption.

Fig. 4c: Replace lowercase letters (i.e. a, b) by capital letters (i.e. A, B).

**Response:** As suggested, we replaced lowercase letters (i.e. a, b) by capital letters (i.e. A, B) in Fig 4c.

Fig. 5a: The vertical axis labels of Fig. 3a and c and the horizontal axis labels of Fig. 5a are written in bold type and they should be changed.

**Response:** Thanks for the observation. We have corrected the vertical axis labels of Fig. 3a and c and the horizontal axis labels of Fig. 5a.

Fig. 6: Explain what the width of arrows means.

**Response:** We thank the reviewer for pointing this out. The width of arrows is proportional to the strength of the relationship, which has now been explained in the figure caption.

Fig. S1 and 2: Replace "N0" and "N10" by "Without N" and "N addition." The terms "N0" and "N10" are not used in the main manuscript.

**Response:** Thanks, we have replaced "N0" and "N10" with "Without N" and "N addition" in Fig. S1 and Fig. S2.

I recommend that authors use consistent color for "Without N" and "N addition" in figures for easy understand. In current manuscript, white bar represents "Without N" in the bar graph, whereas white dot represents "N addition" in the scatter plots. Red fitted line represents "Without N" in Fig. 1c, d and "N addition" in Fig. 2, 3.

**Response:** Thanks for pointing this out. We have corrected the dot figures for consistency using white and black colors to represent "Without N" and "N addition", respectively. Moreover, black and red lines were fitted for "Without N" and "N addition" treatments, respectively.

---

## Author Comment (AC2) · 18 Aug 2019

We sincerely thank the reviewer for the constructive comments and suggestions, which helped us to substantially improve our manuscript. Please find the point-to-point responses (blue) to the comments (black) as listed below.

**Reviewer 2**

P2 Line 14-15: logic leaps exist in the statement 'Moderate-cycling IP was mainly regulated by aboveground plant biomass with $KH_2PO_4$ addition, while by soil pH and plant biomass with addition of $Ca(H_2PO_4)_2$'. Please change 'plant biomass' into 'plant P uptake'.

Response: As suggested, we have reworded "plant biomass" into "plant P uptake" and changed the parameter when re-running the SEM model (P. 2 Line 14).

P2 Line 24: Please change the keyword 'soil phosphorus' into 'nitrogen enrichment'.

Response: As suggested, we have changed the keyword "soil phosphorus" into "nitrogen enrichment" (P. 3 Line 1).

P3 Line 3: change '50-90%' into '50%-90%'.

Response: As suggested, we have changed "50-90%" into "50%-90%" (P. 4 Line 3).

P3 Line 17-18: 'labile/ available P (Olsen P), moderate-cycling P fractions (Al-P, Fe-P, $Ca_2$-P and $Ca_8$-P)'. according to the Methods introduced by the authors, 'Olsen-P was extracted from air-dried soil with 0.5 M $NaHCO_3$ (pH 8.5)(Olsen et al., 1954)', while '$Ca_2$-P was determined by shaking 0.5 g soil with 25 ml 0.25 M $NaHCO_3$ (pH 7.5)' , and in contrast to Fig. 3a,b and Fig5.a,b, $Ca_2$-P is more suitable to classified into 'labile' fraction. The statement should be 'labile P ($Ca_2$-P)/ available P (Olsen P), moderate-cycling P fractions (Al-P, Fe-P, and $Ca_8$-P)'.

Response: Thanks for the comments. We fully agree with the comment and define $Ca_2$-P as labile P in the study. The statement has been changed into "labile P ($Ca_2$-P) / available P (Olsen-P), moderate-cycling P fractions (Al-P, Fe-P and $Ca_8$-P)" (P. 4 Line 17).

P3 Line 21: delete 'understanding'.

Response: Yes, we deleted "understanding".

P4 Line 1: reword 'content' into 'concentration'.

Response: We have replaced 'content' into 'concentration' (P. 5 Line 7)

P5 Line 6: suggest to rephrase 'release of P from soil organic P' into 'mineralization of soil organic P'.

Response: As suggested, we have replaced 'release of P from soil organic P' into 'mineralization of soil organic P' (P. 6 Line 12).

P5 Line 25: 'soil labile IP (Olsen-P)': 'Soil labile IP ($Ca_2$-P) or available P (Olsen-P)'.

Response: As suggested, we have replaced 'soil labile IP (Olsen-P)' into 'soil labile IP ($Ca_2$-P)' (P. 7 Line 6).

P6 Line 1: remove '$Ca_2$-P'.

Response: We have removed $Ca_2$-P in the sentence (P. 7 Line 7).

P7 Line 12-13: maybe, this sentence should be written as 'The plants were sorted to species and oven-dried at 65 for 48 h, then weighed and ground'? Because the way you did these should be to determined aboveground net primary productivity and get plant samples ground to measure plant P concentration.

Response: As suggested, we have changed the description into 'The plants were sorted to species and oven-dried at 65 for 48 h, then weighed and ground' (P. 8 Lines 22-23).

P7 Line 24: change 0.5g into 0.5 g.

Response: As suggested, 0.5g were changed into 0.5 g.

P8 Line 16-18: change 'Total IP (TIP) concentration was defined as the sum of moderate-cycling IP (Al-P, Fe-P, $Ca_2$-P and $Ca_8$-P) and recalcitrant IP ($Ca_{10}$-P and O-P).' into 'Total IP (TIP) concentration was defined as the sum of labile IP ($Ca_2$-P), moderate-cycling IP (Al-P, Fe-P and $Ca_8$-P) and recalcitrant IP ($Ca_{10}$-P and O-P).'

Response: Thanks for the suggestion. We have changed 'Total IP (TIP) concentration was defined as the sum of moderate-cycling IP (Al-P, Fe-P, $Ca_2$-P and $Ca_8$-P) and recalcitrant IP ($Ca_{10}$-P and O-P).' into 'Soil total IP (TIP) concentration was defined as the sum of soil labile IP ($Ca_2$-P), moderate-cycling IP (Al-P, Fe-P and $Ca_8$-P) and recalcitrant IP ($Ca_{10}$-P and O-P)' (P. 9 Line 24-P. 10 Line 1).

P9 Line 24: plant P uptake should be incorporated in the SEM model instead of plant biomass. If plant biomass is included, it should be explained by P fractions/ P availability. But do not use plant biomass to explain P fractions.

Response: As per suggestion, we used P uptake instead of plant biomass in the SEM and reanalyzed the SEM.

P 12 Line 12: Even though plant biomass production consume soil P, it would be better to directly use the parameter of plant P uptake as you calculated in P 9 Line 5.

Response: As per suggestion, we changed "plant biomass" into "plant P uptake" (P. 14 Line 2).

P15 Line 16-17: In this study, you were unable to determine the relative transformation rates of the two compounds. But the results of higher O-P and Ca10-P suggested that a higher proportion of $Ca(H_2PO_4)_2$ was converted into stable inorganic P forms. So, please correct this point.

Response: Thanks for the comment. We have corrected the sentence into "This suggests that a higher proportion of $Ca(H_2PO_4)_2$ was converted into stable inorganic P forms than the more soluble $KH_2PO_4$ (P. 17 Lines 7-9).

P16 Line 7 & 22: Be aware that the mechanistic description might change if you replace the parameter of plant biomass with plant P uptake.

Response: Thanks for the reviewer's observation. We have updated the description in the Result and Discussion section.

P18 Line 19-20 change 'Phosphate Release Kinetics in Calcareous Grassland and Forest Soils in Response to H+ Addition' into 'Phosphate release kinetics in

calcareous grassland and forest soils in response to H$^+$ addition'

Response: Yes, we have corrected 'Phosphate Release Kinetics in Calcareous Grassland and Forest Soils in Response to H+ Addition' into 'Phosphate release kinetics in calcareous grassland and forest soils in response to H$^+$ addition'

P19 Line 9: *Lolium perenne* and *Pinus radiata*, italic.

Response: Yes, we changed Lolium perenne and Pinus radiata into *Lolium perenne* and *Pinus radiate*.

P20 Line 1-2: change 'Do Nutrient Limitation Patterns Shift from Nitrogen Toward Phosphorus with Increasing Nitrogen Deposition Across the Northeastern United States?' into 'Do nutrient limitation patterns shift from nitrogen toward phosphorus with increasing nitrogen deposition across the Northeastern United States?'

Response: This has been corrected.

P20 Line 24-25: change 'Soil fertility and fertilizers an introduction to nutrient management' into 'Soil Fertility and Fertilizers: An Introduction to Nutrient Management'

Response: Yes, we have changed 'Soil fertility and fertilizers an introduction to nutrient management' into 'Soil Fertility and Fertilizers: An Introduction to Nutrient Management'.

P24 Line 20-21: change 'Nitrogen Fertilization Effects on Grassland Soil Acidification: Consequences on Diffusive Phosphorus Ions' into 'Nitrogen fertilization effects on grassland soil acidification: consequences on diffusive phosphorus ions'

Response: Yes, we have changed 'Nitrogen Fertilization Effects on Grassland Soil Acidification: Consequences on Diffusive Phosphorus Ions' into 'Nitrogen fertilization effects on grassland soil acidification: consequences on diffusive phosphorus ions'.

P26 Line 1: 'Larix gmelinii', italic?

Response: Yes, we changed 'Larix gmelinii' into '*Larix gmelinii*'.

P30 Line 8: 'P', italic.

Response: Yes, we changed "P" into "*P*".

P33 Line 8: 'P', italic.

Response: Yes, we changed "P" into "*P*".

P35 Fig. 6: reshape the 'Structural equation model' and rewrite the Results and Discussion sections related to the SEM.

Response: As suggested, we have reshaped the SEM and rewrote the Results and Discussion sections related to the SEM (P. 13 Line 23-P. 14 Line 3, P. 16 Lines 19-20, P. 17 Lines 20-24).

P36 Fig. 7: The content in the figure is hard to read because of the background color. Remove $Ca_2$-P from the 'moderate-cycling P', and instead of using Olsen as labile P, I suggest the authors use $Ca_2$-P as labile P.

Response: As suggested, we have changed the background color in Fig. 7. We removed $Ca_2$-P from the 'moderate-cycling IP' and use $Ca_2$-P as labile IP.

---

## Author Response (AR2)

Dear Professor Nobuhito Ohte,

Thank you for sending us the comments and we appreciate the review work from you and two anonymous reviewers. Here, we have revised the manuscript according to reviewer's comments and suggestions and listed our point-by-point responses below. We hope that you will find the new version of our manuscript is now suitable for publication in Biogeosciences. Reference to line numbers is for the version without trace changes.

**Reviewer #1**

p. 11 line 16: I do not think that Nitrogen addition always significantly increased aboveground biomass production for within each P addition rate (Some bars have no asterisks). Eliminate "significantly" or add "xx kg P ha$^{-1}$ yr$^{-1}$".

**Response:** As suggested, "significantly" has been eliminated (P11 Line 16).

p. 11 line 22-23: According to results of multiple comparison in Fig. 1f, soil pH does not decrease without N addition on $Ca(H_2PO_4)_2$ treatment.

**Response:** We thank the reviewer for the comment. Now, this sentence has been rephrased into "$Ca(H_2PO_4)_2$ tended to decrease soil pH at 80 and 100 kg P ha$^{-1}$ yr$^{-1}$ without N addition and at 60, 80 and 100 kg P ha$^{-1}$ yr$^{-1}$ with N addition" (P11 Lines 23-24).

p.12 line 20-23: Table S2 does not show the relevant result. As far as I see Table S2, I think that description "40, 60 and 100 kg P ha$^{-1}$ yr$^{-1}$ with N addition" should be changed to "20, 60 and 80 kg P ha$^{-1}$ yr$^{-1}$ with N addition". Reconfirm your dataset and description in results.

**Response:** The dataset has been reconfirmed. We agree with the reviewer that the description should be changed to "20, 60 and 80 kg P ha$^{-1}$ yr$^{-1}$ with N addition" (P12 Line 23). Because the comparison of $Ca_{10}$-P concentration was made between $KH_2PO_4$ and $Ca(H_2PO_4)_2$ treatment at the same P level with and without N separately. The *P* value from the comparison has been provided at the right side which showed insignificant results at 20 ($P = 0.09$), 60 ($P = 0.05$) and 80 P ha$^{-1}$ yr$^{-1}$ ($P = 0.18$) with N addition.

Technical corrections
p. 4 line 23, p. 10 line 10, p. 16 line 22 and line 24, p. 19 line 13: Olsen P => Olsen-P

**Response:** We thank the reviewer for the observation. These technical issues have been corrected.

**Reviewer #2**

P14 Line 25: change 'then' into 'eventually'

**Response:** This has been corrected (P15 Line 1).

P17 Line 13: 'in the grassland soil'

**Response:** It has been corrected (P17 Line 14).

P18 Line 5: change 'semi-arid' into 'meadow'

**Response:** Thanks for the observation. It has been corrected (P18 Line 6).

[revised manuscript text omitted]